# Machine Learning for Brain MRI Data Harmonisation: A Systematic Review

**DOI:** 10.3390/bioengineering10040397

**Published:** 2023-03-23

**Authors:** Grace Wen, Vickie Shim, Samantha Jane Holdsworth, Justin Fernandez, Miao Qiao, Nikola Kasabov, Alan Wang

**Affiliations:** 1Auckland Bioengineering Institute, University of Auckland, Auckland 1142, New Zealand; 2Centre for Brain Research, University of Auckland, Auckland 1142, New Zealand; 3Mātai Medical Research Institute, Tairāwhiti-Gisborne 4010, New Zealand; 4Department of Anatomy & Medical Imaging, Faculty of Medical and Health Sciences, University of Auckland, Auckland 1142, New Zealand; 5Department of Computer Science, University of Auckland, Auckland 1142, New Zealand; 6Knowledge Engineering and Discovery Research Institute, Auckland University of Technology, Auckland 1010, New Zealand; 7Intelligent Systems Research Centre, Ulster University, Londonderry BT52 1SA, UK; 8Institute for Information and Communication Technologies, Bulgarian Academy of Sciences, 1113 Sofia, Bulgaria

**Keywords:** systematic review, harmonisation, normalisation, standardisation, MRI, image pre-processing

## Abstract

Background: Magnetic Resonance Imaging (MRI) data collected from multiple centres can be heterogeneous due to factors such as the scanner used and the site location. To reduce this heterogeneity, the data needs to be harmonised. In recent years, machine learning (ML) has been used to solve different types of problems related to MRI data, showing great promise. Objective: This study explores how well various ML algorithms perform in harmonising MRI data, both implicitly and explicitly, by summarising the findings in relevant peer-reviewed articles. Furthermore, it provides guidelines for the use of current methods and identifies potential future research directions. Method: This review covers articles published through PubMed, Web of Science, and IEEE databases through June 2022. Data from studies were analysed based on the criteria of Preferred Reporting Items for Systematic Reviews and Meta-Analyses (PRISMA). Quality assessment questions were derived to assess the quality of the included publications. Results: a total of 41 articles published between 2015 and 2022 were identified and analysed. In the review, MRI data has been found to be harmonised either in an implicit (*n* = 21) or an explicit (*n* = 20) way. Three MRI modalities were identified: structural MRI (*n* = 28), diffusion MRI (*n* = 7) and functional MRI (*n* = 6). Conclusion: Various ML techniques have been employed to harmonise different types of MRI data. There is currently a lack of consistent evaluation methods and metrics used across studies, and it is recommended that the issue be addressed in future studies. Harmonisation of MRI data using ML shows promises in improving performance for ML downstream tasks, while caution should be exercised when using ML-harmonised data for direct interpretation.

## 1. Introduction

Neuroimaging technologies have rapidly developed in recent years. They are now widely used in various neurological research centres to study various clinical syndromes [1,2,3]. To both reduce logistics costs and increase the diversity of clinical exemplars, large-scale neuroimaging studies often combine datasets from multiple centres [4,5]. The presence of heterogeneity in Magnetic Resonance Imaging (MRI) machine characteristics, such as variations in software, MRI equipment, reconstruction algorithms and scan protocols, can lead to inter- and intra-site variations. Such variations can affect the signal intensity and derived metrics, which may obscure the effect of interest [6,7,8,9,10,11] and lead to the failure of downstream and machine learning (ML) analyses [12,13].

To address these issues, harmonisation can be applied to remove non-biological factors that can cause variations in MRI data across different centres and institutions. This can be achieved through applying protocol or data harmonisation. An acquisition protocol is harmonised by specifying the type of MRI machine, as well as the imaging parameters, e.g., the field of view and the analysis methods [14]. However, using standardised acquisition protocols is not sufficient to achieve the desired level of standardisation and cannot be applied to existing datasets [15,16,17]. In contrast, data harmonisation aims to improve the comparability of MRI data so that scan results are comparable across different settings/domains (e.g., sites, scanners, and institutions) [18]. The underlying objective is to remove non-biological factors while preserving biological factors among participants from different domains [18]. This differs from data normalisation, which compares and integrates MRI data across subjects within the same domain by transforming the data into a standardised system. Harmonisation is performed across different domains.

Data harmonisation can be challenging due to the presence of confounding variables and unknown factors that cause cross-site variations. Direct mapping to a constant value may not be sufficient because it could eliminate important variables of interest. The use of ML, especially deep learning, to address the scanner effect has grown significantly due to its ability to model complex relationships within messy data sets. Using ML techniques, data can be harmonised either explicitly or implicitly. Explicit approaches aim to predict the target image or its derived metrics given the reference image. ML models are trained to harmonise data which can then be directly inspected by a radiologist. Models used in an implicit manner, on the other hand, are trained on a downstream task (e.g., classification and segmentation), and harmonisation is performed implicitly through the optimisation process during training. Therefore, the harmonised data are not visible. A representation of explicit and implicit methods for data can be seen in Figure 1.

This paper presents a systematic review of harmonisation methods for brain MRI data, with a specific focus on ML-based approaches applied in both explicit and implicit ways. Our primary objective is to provide an overview of the latest techniques and advancements in MRI harmonisation that will be valuable for radiologists, neuroimaging researchers, and other professionals working with large-scale, multicentre MRI datasets. While several reviews have been conducted to summarise the harmonisation methods used in medical imaging [18,19,20], none have focused on ML-based approaches. The field also lacks a summary of implicit methods. Therefore, this review aims to fill these gaps and offer valuable insights into available resources when developing harmonisation methods.

## 2. Materials and Methods

The Preferred Reporting Items for Systematic Reviews and Meta-Analyses (PRISMA) guidelines were followed in conducting this review. Our main interest is: What ML algorithms are available, and how are they being applied by researchers to harmonise brain MRI data.

### 2.1. Inclusion Criteria and Search Terms

To focus on our main question of interest, we reviewed works that meet the following inclusion criteria: (1) address the multi-site/scanner/centre issues by using data harmonisation methods; (2) apply ML techniques; and (3) are applied to MRI images. The search terms were generated by considering each inclusion criterion individually. For the first criterion, we used terms such as “harmonisation”, “normalisation”, and “standardisation”, as well as those associated with multicentre, such as “multi-site”, “inter-site”, and “multi-scanner”. For the second criterion, we used a comprehensive range of search terms encompassing ML algorithms that had used or could potentially be used for harmonisation. The third criterion involved keywords related to MRI, such as “DTI”, “MRI”, “DWI”, and “functional”. All keywords used for searching is shown in Table 1. To form the sophisticated search terms, the Boolean expression ‘OR’ was used to combine within-group keywords, while ‘AND’ was used to combine three groups. Parenthetically, articles from both journals and conference proceedings have been included in this review since they both make significant contributions to the fields of ML and bioengineering.

### 2.2. Screening and Selection Process

Articles published in journal articles or conference proceedings between Jan 2010 and June 2022 were searched using three search engines: PubMed, IEEE, and Web of Science. Following the PRISMA, the first phase of screening removed duplicate articles collected from multiple sources. In the second phase of screening, we discarded all papers that conclusively did not belong in the scope of this review, such as those with abstracts and titles which contained no keywords related to “harmonisation” or “machine learning”. Afterwards, the articles that passed the screening phase were assessed for eligibility by reviewing the full texts. In addition to the usual search engines, papers identified from the reference list of included papers were also evaluated for eligibility using the same criteria. Searching and screening were conducted independently by two researchers (G.W and A.W).

### 2.3. Exclusion Criteria

Articles that were excluded from the review include those that (a) were not peer-reviewed; (b) were books, letters, notes, graduate theses, and patents; (c) were not able to be accessed; (d) were not written in English; (e) were surveys or literature reviews; (f) did not use multi-site or multi-scanner datasets; (g) were not applied on human neuro MRI; (h) did not sufficiently describe an ML approach or an experiment or validation methods; (i) did not contain any quantitative results; or (j) did not describe an experiment or validation study.

### 2.4. Data Extraction

From the data, we extracted: the author name, publishing details, data set details, harmonised domain, training inputs and validation strategies, evaluation methods, computational libraries, ML techniques and study design. More details about the data characteristics are presented in Table 2. Extracted data were evaluated and resolved (if necessary) by another reviewer (V.S.).

### 2.5. Quality Assessment

The modified QualSyst Assessment Tool for quantitative studies was used to assess the methodological quality, credibility and relevance of the included studies [21]. The assessment involved the use of questions that were derived from the scale, focusing on aspects such as research questions, study design, data collection, data analysis, and results reporting. Additionally, questions related to the quality of ML-based models, such as the ML algorithms, training procedure and comparative analysis, were incorporated, as suggested by [22]. These questions are listed in Table 3. Each of them had three optional answers: “Yes”, “Partly”, or “No”, with scores of 1, 0.5, or 0, respectively. For each study, the final score was calculated by summing up the scores given to each question, ranging from 0 to 10. A cut-off score of 5 (50% of the maximum total) was considered acceptable when assessing the reliability of studies. Final scoring was agreed upon by two reviewers (G.W. and A.W.).

## 3. Results

Figure 2 shows the PRISMA flowchart. We identified a total of 1112 studies from three different bibliographic databases, as well as six additional studies from other sources. After removing duplicates, 1053 studies were screened based on their titles and abstracts, resulting in 97 studies for full-text evaluation. Eventually, 41 articles published between 2015 and June 2022 remained for quality assessment. The quality scores for the included studies, ranging from 5.5 to 10 with a mean score of 8.16, are provided in Appendix A. These findings suggest that the studies included in this review are of high quality. Scores for questions 8 and 9 were relatively low, however, indicating that several studies did not conduct adequate comparative analyses. Specifically, 24% of the studies did not perform any comparative analysis, and 48.8% of the studies compared their proposed method to either statistical or ML-based approaches. The results also indicated a slight advantage in scoring for studies utilising implicit approaches compared to those utilising explicit approaches. However, the sample size was relatively small, and subject characteristics were unclear, contributing to bias. A table with extensive details for individual key aspects can be seen in Appendix A.

### 3.1. Article Distribution among Journals and Conference Proceedings

The eligible articles chosen for the study were published across sixteen different journals and three conference proceedings. In total, 29 papers were published in journals, and 12 papers were from conference proceedings. Among these, *the Medical Image Computing and Computer Assisted Intervention Society* conference proceedings contained the most publications used, comprising 9 out of 41 publications. Following this, six publications from *NeuroImage* and five from *Medical Image Analysis*. The publications covered a diverse range of journals, including those in the fields of biomedical engineering, neuroscience, signal processing, and computer science. Figure 3 depicts all the included publishing sources, journals, and conference proceedings.

### 3.2. Application of ML Methods

A total of 41 studies were conducted that describe ML being applied to harmonise MRI data in either an explicit (*n* = 20) or implicit (*n* = 21) way. The applied ML algorithms were plotted by year, shown in Figure 4. Implicit approaches first appeared in the included studies in 2015, while explicit harmonisation first appeared in 2019. Generally, the number of papers has grown gradually since 2019. Explicit approaches included generative adversarial networks (GANs) (*n* = 7), autoencoders (*n* = 7), convolutional neural networks (CNNs) (*n* = 3), regression (*n* = 1), transformer networks (*n* = 1) and sparse dictionary learning (*n* = 1). Regarding implicit approaches, most of the applications were fine-tuning (*n* = 10), followed by adversarial transfer learning (*n* = 7), feature extraction network-based transfer learning (*n* = 3) and multi-task learning (*n* = 1). These ML methods are presented in Table 4, along with a brief description and the number of articles for each method.

### 3.3. MRI Modality and Harmonised Features

Three different MRI modalities were identified: structural MRI (sMRI) (*n* = 28), diffusion MRI (dMRI) (*n* = 7), and functional MRI (fMRI) (*n* = 6). Papers were also categorised according to the harmonisation features: raw signals (*n* = 29) and image-derived features (*n* = 12). Of the 28 studies on sMRI, 24 harmonised raw intensity signals, while four studies focused on derived measures, including brain volumes [25,46], cortical thickness [36] and mixed image-based features [66]. Four studies harmonised the raw diffusion signals of dMRI through dictionary representation [31] and spherical harmonics representation [29,39,48], while three studies focused on derived features [28,41,65]. For fMRI data, only one study harmonised raw signals [61], whereas five harmonised derived functional connectome-based features [57,60,62,70,73]. Results showed that raw signals of sMRI and dMRI data were more frequently used than fMRI raw signals. One possible explanation for this is that fMRI data are time-series signals, which have a higher degree of dimensionality and are, therefore, more complex and difficult to process and analyse.

### 3.4. Evaluation of Methodology

The reliability of harmonised data can be evaluated directly (*n* = 20) and indirectly (*n* = 25), depending on whether travelling subject datasets or downstream ML models were available. Direct evaluation methods were more widely used in explicit approaches. One method was to evaluate the appearance of the harmonisation results. The contrast of harmonised images was found to be more similar to that of reference scans, as documented in studies such as [27]. Studies also performed subject-wise comparisons by measuring similarities between individual scans from different domains. This was accomplished using various metrics such as mean error (in studies, e.g., [27,28,36]), structural similarity index (in studies, e.g., [27,35,49]) and coefficient of variation (in studies, e.g., [28,48]).

Alternatively, a group-wise analysis can be performed to determine whether subjects between different groups, such as different age groups [25], become more distinguishable after harmonisation. Studies that used explicit methods commonly employed statistical tests to compute the probabilities of differences. The most commonly used metrics were Cohen’s d value (e.g., [36,41]), Pearson’s r value (e.g., [41]), and Kullback–Leibler divergence (e.g., [31,46]). Implicit methods, on the other hand, used the t-SNE algorithm to directly visualise the data distributions based on the features extracted from the trained network. Three studies have used this method, which revealed a significant change in domain clustering after harmonisation [51,52,71].

Using indirect evaluation methods, researchers set up the downstream task and measured the harmonisation effect by comparing task performance before and after harmonisation. Implicit methods relied solely on indirect evaluation methods, while only four studies demonstrated the harmonised results using downstream tasks in explicit methods [33,35,37,40].

There were a variety of approaches taken to validate the models developed. Validation by splitting into a testing and validation dataset was performed in all studies. In total, 22 studies used k-fold cross-validation.

### 3.5. Data Characteristics and Reproducibility

MRI data is typically stored in a 3D or 4D format. Three categorises were identified: 1D (*n* = 6), 2D (*n* = 19), pseudo 3D (*n* = 3) which is constructed by 2D data, and 3D (*n* = 13). We also documented the accessibility of the datasets, the number of subjects, and the availability of travel subjects. Thirty papers in our review used public datasets that are available for download, eight studies used private datasets (*n* = 8), and three studies used both. Thirteen studies utilised travelling subject datasets. In total, nine datasets containing travelling subjects were found, as shown in Table 5. Five of them are publicly available.

We divided the articles into five categories based on the size of the datasets used. The number of studies in each category is shown in Figure 5. Most of the studies used datasets with sizes of 101–1000 subjects (*n* = 20), followed by 1–20 subjects (*n* = 7), 1001–10,000 subjects (*n* = 8), 21–100 subjects (*n* = 5) and more than 100,000 subjects (*n* = 1). In terms of reproducibility, less than 30% of the studies (*n* = 11) reported their implemented codes. Of the 60% of studies (*n* = 23) that described the libraries used for implementation, PyTorch (*n* = 11) was the most widely used library, followed by Keras and TensorFlow (*n* = 6).

## 4. Discussion

This systematic review evaluated the use of ML-based algorithms for the purpose of MRI data harmonisation. We summarise the main findings in methodology, datasets, and evaluation methods, highlighting their limitations and offering suggestions for future directions.

### 4.1. ML-Based Methods for Harmonisation

ML methods have provided efficient solutions for explicitly generating harmonised MRI data across different domains. The most widely used models are GANs and autoecoders, which have shown promising results in reducing multi-site variation through image-to-image synthesis [40,46,49]. GANs perform domain translation by learning domain-invariant features [34,80]. One issue is how they are limited to mapping between two specific scanners for most studies [36,37,38,39,40,41]. When mapping between multiple sites, multiple generative models are required, increasing the difficulty of training [35]. Another issue is that unpaired image-to-image translations may not preserve heterogeneity and individual quantitative information [81,82].

Autoencoder-based methods aim to harmonise data in terms of disentangled representations [83]. All related work in this review attempted to extract scanner-related features for harmonisation [43,44,45,46,47,48,49]. Similar to GANs, data from multiple sources and target domains are required for training. However, when an additional dataset from a new scanner is given, re-training only needs to be conducted on the new dataset. It is worth noting that none of the studies examined the minimum number of subjects required to disentangle the representation during training. Future study is needed to address this gap in knowledge.

Implicit approaches focus on finding task-specific, scanner-related confounds using either feature- or model-based techniques. Feature-based techniques, including feature extractors and adversarial networks, perform distribution alignment. This is, however, challenging, especially when optimising adversarial objectives due to the complex feature space of 3D or 4D MRI data. While most studies focused on alignment between two specific domains, one study attempted to create a global shared domain, resulting in a more generalisable and transferable feature representation that can be applied to new domains [61].

Fine-tuning and multi-task learning are model-based transfer learning techniques [74,84]. One future direction for these methods is to investigate the optimal number of subjects required for model transferring [85]. Research should also explore the adaptability of models to unseen domains.

### 4.2. Dataset Used for Harmonisation

Travelling subject datasets are essential for explicit harmonisation, as they can ensure that the ML models do not learn the population bias [10,46,86]. Current travelling subject datasets, however, have limitations due to their small size, the long inter-scan interval between consecutive scans of the same individual and scan-rescan reliability issues. Most datasets contain no more than 20 travelling subjects, and while a few are relatively large, they are primarily designed for longitudinal study for capturing brain changes over time. Addressing scanner-related problems can be difficult when the biological features of the brain differ significantly between the paired scans. Scan-rescan reliability can be influenced by factors such as positioning, shim setting and some unknown factors [87,88]. These variabilities are usually neglected in a study, but they can be significant, particularly given the small size of the travelling subject dataset.

Implicit approaches or unsupervised learning do not require a travelling-subject dataset. Instead, researchers need to define the source and reference domains. Large-scale public datasets may not, however, provide enough information, such as site information regarding where scans were acquired from, making harmonisation challenging. One solution is to simplify the problem by performing mapping between broader domains, for example, between different field strengths (e.g., 1.5 T, 3.0 T). Studies have shown that harmonising data within broader domains can reduce non-biological variations [33,47]; however, whether this is sufficient has yet to be explored.

As the number of public datasets grows, ML models can be validated more effectively against more diverse datasets. While there are shared travelling subject dataset collections, many have not been recently updated. One example is the multiBrain collection (https://github.com/Conxz/multiBrain, accessed on 15 February 2023), which has not been updated since 2019. Additionally, we identified the RMP Rumination fMRI Dataset, which is a travelling subject dataset that has not been used in any of our included studies [89]. It is plausible that this dataset has been used in low-quality studies that were not included in this review. Further evaluation is necessary to determine the efficacy of this dataset for harmonisation. To address the growing problem of scanner-related issues, an open and maintained platform for sharing travelling subject datasets should be developed and established in the future. Public data sources should include site and scanner information, being as specific as possible.

### 4.3. Data Representation for Model Training

Most ML models in this review only used 2D inputs. In general, 3D models are more powerful for exploring medical image features and yield better learning performance [90]. Developing models that capture 3D and 4D structural information is, however, challenging due to subtle scanner differences. Current 3D models primarily use patch-based architectures due to limited computational resources. Nevertheless, how to improve the state-of-the-art patch-based networks to integrate location information for harmonisation is still an open research question. Possible solutions could be to incorporate brain location information into a patch-based model, as suggested in [91], or to use a multi-scale patch-based network, as demonstrated in [92]. In addition, 4D models have been introduced in the field of medical images [93], but their applications in harmonisation are lacking. Developing 4D models to harmonise fMRI data or introduce domain adaption techniques in 4D models would be a promising future direction.

### 4.4. Performance Evaluation for Harmonisation Models

Ideally, harmonisation should remove all non-biological factors while preserving biological factors. Since there is no reference for harmonisation, methods evaluation can be difficult due to unknown sources of variability. Evaluation methods for explicit harmonisation methods have been performed using various quantitative methods, but heterogeneity in evaluation methods across studies makes it difficult to synthesise results quantitatively or conduct meta-analyses. Implicit approaches are simpler to evaluate by comparing prediction accuracy before and after harmonisation.

Image quality assessment (IQA) is also important for the evaluation of image-to-image tasks. Studies utilising explicit harmonisation methods have shown that applying ML-based harmonisation can enhance image quality, as demonstrated using full-reference IQA metrics such as signal-to-noise ratio and the contrast-to-noise ratio [35,40,43,44,49]. No-reference IQA methods, e.g., blind/referenceless image spatial quality evaluator [94], are, however, lacking.

It is recommended that researchers use more advanced IQA methods, such as in [95] or ML-based methods, such as in [96,97]. A reliable and robust reference-free IQA framework for harmonisation should be developed and used in future studies. Such a framework should be capable of evaluating different MRI modalities and different sequential stages (e.g., slice-wise, volume-wise and subject-wise) [98]. In addition to evaluating whether the quality of the harmonised image has been improved, it is also important to assess whether the criteria have been met, as well as the trade-off between the computational cost and the improved quality. A thorough quality check is essential before utilising harmonised images for any analysis.

### 4.5. Comparative Analysis

For explicit approaches, studies often compare their proposed methods to either statistical methods or other state-of-the-art ML methods. In statistical comparative analysis experiments, studies consistently suggest that their proposed ML-based methods outperform the statistical features alignment methods [28,41,44,44,45,46,47,49]. When comparing different ML methods, some studies suggest that autoencoders outperform GANs [44,49], while others find that supervised CNNs models outperform Encoder-based methods [27] and transformer networks outperform GANs [33]. Some studies also show improvements over previous, similar studies [31,48].

For implicit approaches, studies have consistently demonstrated that incorporating harmonisation leads to better performance compared to baseline experiments which do not apply harmonisation. Transfer learning approaches tend to outperform statistical covariate techniques [50,52,55,56,59,60,61,63,64,65,69,71,74], and some studies suggest that adversarial transfer learning methods outperform non-adversarial methods [51,53,55]. Multi-task learning is superior to single-task learning in classification tasks [75]. Some studies also demonstrate improvements over various feature-based methods when using their proposed methods [57,60,61,62,69,72]. It is important to note that studies that perform different regularisation techniques are not counted as comparative analyses in this study.

Although several studies have attempted to compare different models using the same dataset [99,100], no conclusions regarding whether high-performing ML models are necessary for MR data harmonisation can be drawn. This review does not include these studies as they do not fall within the scope of developing new methods. These studies, however, highlight the importance of carefully evaluating and comparing different methods to determine their effectiveness for a given task and dataset.

### 4.6. Toward Clinical Application of AI

The application of AI in clinical settings is an exciting prospect. Harmonisation can improve the accuracy and generalisability of AI models, allowing for more accurate results and better generalising to new datasets. To further enhance personalised, predictive and explainable computational models, new ML methods, such as brain-inspired computation [101], are worth exploring, along with traditional methods. An example of using a brain-inspired approach on longitudinal sMRI data is presented in [102]. In [103], fMRI and DTI neuroimaging data are integrated with a spiking neural network model based on the NeuCube [104,105] architecture for a better personalised and explainable prediction of response to treatment of Schizophrenia patients.

The ideal harmonisation method that can be applied in a clinical setting should be capable of removing variable data caused by using multiple scanners without requiring the testing data to be drawn from the same group as the training data. This, however, remains an ongoing challenge.

### 4.7. Recommendation for Future Studies

To harmonise several MRI datasets, the choice of ML methods should depend on the specific research goals and constraints of the datasets. Therefore, we have the following suggestions for researchers:Understanding the purpose of harmonisation. Use explicit approaches for harmonising intensity values and image-derived metrics or correcting known sources, whereas using implicit approaches to improve the performance of a downstream task or to correct unknown sources of variability.Consider the nature and properties of the dataset, including the size, dimensionality, and variability in the data. Generally, the more extensive and diverse the dataset, the more complex the ML model can be. For example, when using a travelling-subject dataset for training, researchers may consider using simpler ML models to reduce the risks of overfitting.When defining the target and reference domains, ensure that there is enough training data in each domain so that the model can learn the relevant information.Conduct experiments using different ML approaches, varying feature extraction techniques, and adjusting hyperparameters. Evaluate the performance using the consistent metrics and interpret the results of image analysis carefully.Consider the trade-off between accuracy and interpretability when choosing an ML method. ML methods, such as GANs, are more complex, so although they may produce highly accurate results, they can be difficult to interpret. In contrast, other methods, such as regression models, may be more interpretable despite giving less accurate results.

In general, we suggest using ML-based harmonisation methods to help improve the performance of the downstream task but to be careful when using ML-harmonised data for direct interpretation. Future studies should focus on developing specific evaluation and IQA methods for ML-harmonised data. In addition, this review focuses on brain MRI, but we believe that the methodologies can be generalised to MRI data of other body parts, such as lungs and breasts [106,107], in the field of harmonisation.

### 4.8. Limitation of Systematic Review

This systematic review has several limitations. First, this review did not evaluate the performance of ML models or compare ML models with other models due to a lack of consistent evaluation metrics. This limitation may hopefully be alleviated by summarising the findings in the comparative experiments conducted in all included studies, as well as the significant usage of the ML models and a broad range of relevant characteristics, such as the evaluation methods. We recommend that future researchers define consistent evaluation methods and metrics for MRI data harmonisation to facilitate comparative analysis for different ML-based applications. Additionally, it is possible that a study may have been overlooked, such as for non-English articles, unpublished or internal studies, or a study that was published after the search procedure was conducted.

## 5. Conclusions

The rapid development of ML in the current big data era necessitates the use of large-scale datasets. It is essential to apply harmonisation when a dataset is heterogeneous with respect to scanner, site, and acquisition parameters. Data harmonisation helps minimise the non-biological variations within a dataset. It can be applied as an essential pre-processing step to minimise scanner-related errors and improve the medical image quality, and potentially improve the performance and generalisability of downstream ML-based prediction models. Harmonisation is therefore important in the medical field, where accurate and consistent diagnoses can have a significant impact on patient outcomes,

This paper performed a systematic review of studies that used ML on MRI data harmonisation for inter-scanner variability removal. We identified and summarised different aspects found in the harmonisation literature, including data sources, modality, features, and ML methods. Our main contribution lies in identifying the current gaps and limitations in the literature and providing recommendations for future research directions. Specifically, we argue in the paper that:-There is a large amount of diverse imaging data (sMRI, dMRI, fMRI, DTI, longitudinal, static etc.) related to the same problem, but collected at different sites under different conditions, that need to be harmonised.-New methods for neuroimage data harmonisation are needed for better results.-Following harmonisation, methods for the integration of the multimodal harmonised data are needed for the development of better (1) personalised, (2) predictive and (3) explainable computational models.-The use of harmonisation as a strategy for improving downstream tasks is recommended; however, consistent evaluation methods are needed.

A growing interest in harmonisation will lead to adopting more advanced ML techniques when developing harmonisation methods. There should be a particular focus on establishing a shared space and creating a universal model for harmonising MRI data, regardless of the scanner, site characteristics, and type of neuroimaging data. The research space should also include some other data modalities.

## Figures and Tables

**Figure 1 bioengineering-10-00397-f001:**
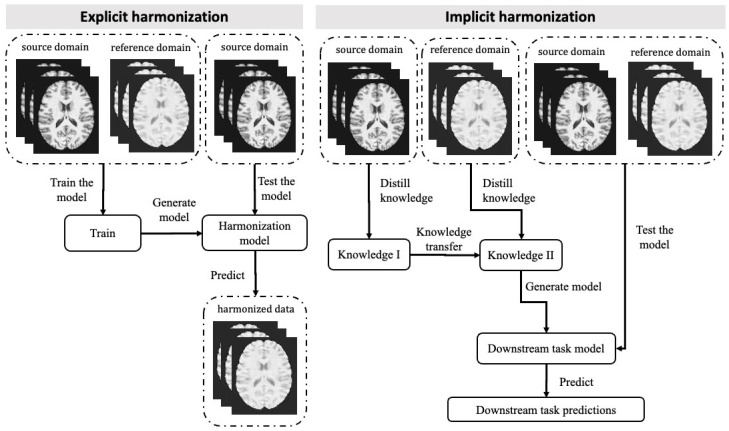
Workflow of Implicit and Explicit Computational Harmonisation Approaches for MRI Data. The explicit approach (**left panel**) involves training data from both source and reference domains to generate the harmonisation model. The source domain refers to the dataset that requires harmonisation, while the reference domain refers to the dataset that serves as a standard. The model is then tested using additional source domain data, resulting in the generation of harmonised data. In the implicit approach (**right panel**), data from the source and reference domains are processed separately, and the distilled knowledge is transferred from one to the other. The downstream task model is generated based on the transferred knowledge, and both source and reference domain data can be fed into the model to generate task-related predictions.

**Figure 2 bioengineering-10-00397-f002:**
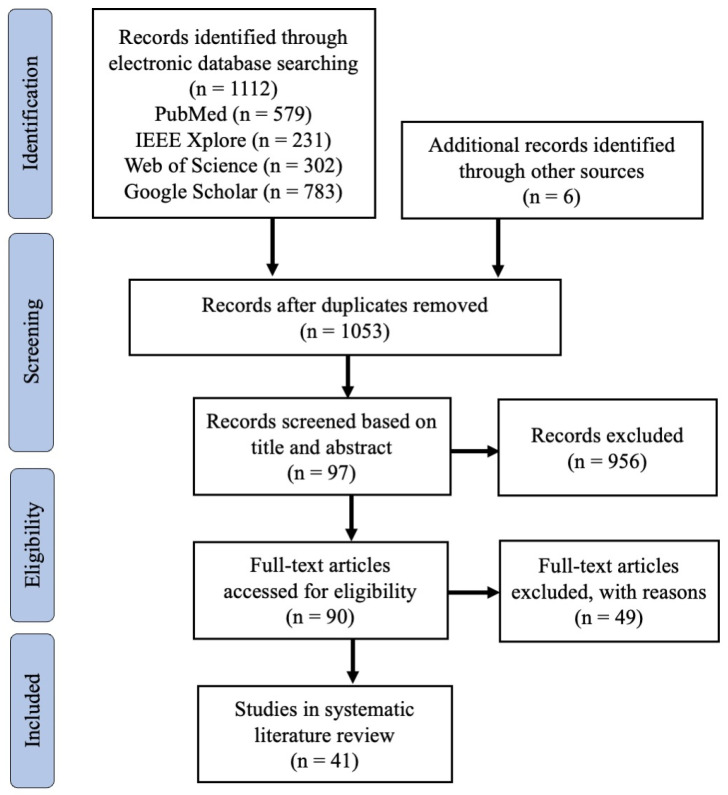
The PRISMA flow diagram for the systematic review detailing the database searches, the number of abstracts screened, and the full texts retrieved.

**Figure 3 bioengineering-10-00397-f003:**
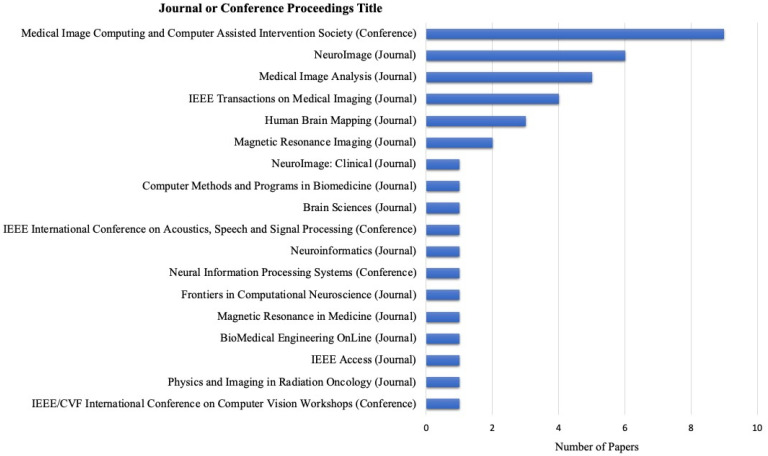
The number of publications per publisher.

**Figure 4 bioengineering-10-00397-f004:**
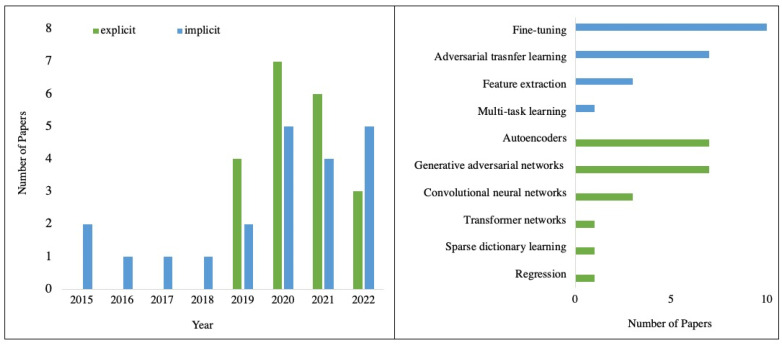
Breakdown of the papers included in this review in the year of publication and harmonisation methods. The number of papers for 2022 has been extrapolated from those published before June.

**Figure 5 bioengineering-10-00397-f005:**
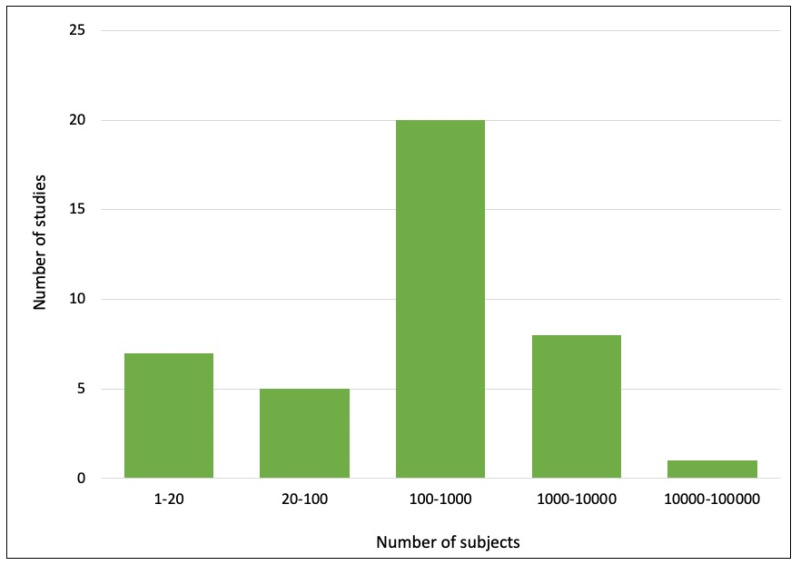
Size of the dataset used in the reviewed studies.

**Table 1 bioengineering-10-00397-t001:** Search terms for electronic databases used.

Categories	Keywords
ML methods	machine learning, Artificial Intelligence, Naive Bayes, Bayesian learning, neural network, neural networks, support vector, random forest, boosting, deep learning, machine intelligence, computational intelligence, ComBat, reinforcement learning, decision tree, linear discriminant analysis, regression, supervised learning, unsupervised learning, independent component analysis, dictionary learning, domain adaptation, generative model
Harmonisation	harmonisation, normalisation, normalisation, multi-site, multicentre, multi-scanner
Brain MRI	MRI, magnetic resonance imaging, imaging, neuroimaging, diffusion MRI, DWI, DTI, structural MRI, T1, T2, proton density-weighted (PD), susceptibility-weighted imaging, fluid-attenuated inversion recovery (FLAIR), double inversion recovery (DIR), functional MRI perfusion MRI

**Table 2 bioengineering-10-00397-t002:** Data characters of the Included studies.

Section	Item	Definitions	Values (Counts If Applicable)
Data	Imaging modality	What is the modality of the input data?	Structural MRI (*n* = 28), diffusion MRI (*n* = 7), functional MRI (*n* = 6)
	Harmonised features	What feature of MRI data is harmonised?	Raw signals (*n* = 29), derived features (*n* = 12)
	Property	Whether the dataset(s) is (are) in-house or public; whether travelling subject dataset(s) (datasets that contain multiple scans of each participant) include(s)?	In-house (*n* = 8), public (*n* = 30), in-house and public (*n* = 3); travelling subjects included (*n* = 13), otherwise (*n* = 28)
Model Selection	Study design	Whether the study aims to produce harmonised data or not?	Implicit approach (*n* = 21), explicit approach (*n* = 20)
	Algorithm selection	What ML algorithm(s) or model(s) is (are) adapted to the proposed methods?	Explicit: random forest regression (*n* = 1), convolutional neural networks (*n* = 3), transformer networks (*n* = 1), generative adversarial networks (*n* = 7), dictionary learning (*n* = 1), autoencoders (*n* = 7). Implicit: adversarial learning (*n* = 7), feature extraction (*n* = 3), fine-tuning (*n* = 10), multi-task learning (*n* = 1)
Model training	Input dimension	What is the dimension of the input data?	1D (*n* = 6), 2D (including 2.5D, *n* = 22), 3D (*n* = 13)
	Input size	What is the input size of the model?	256 × 256, 16 × 16 × 16 etc.
	validation	What validation method(s) is (are) used?	Cross validation (*n* = 23), split dataset (*n* = 18)
Model evaluation	Evaluation approaches	What approaches are used to evaluate the proposed method? Indirect evaluation is when a downstream prediction model is used; otherwise, is direct evaluation?	direct evaluation (*n* = 20), indirect evaluation (*n* = 25)
	Evaluation metrics	What metrics are used to quantitatively evaluate the proposed method?	Structural similarity index, signal-to-noise ratio, etc.
Implementation	Code and reproducibility	Does the author provide the source code?	Yes (*n* = 11), No (*n* = 30)
	Implementation	What programming library is used to build the model?	Keras and/or TensorFlow (*n* = 6), PyTorch (*n* = 11), others (*n* = 6), unknown (*n* = 18)

**Table 3 bioengineering-10-00397-t003:** Quality assessment questions.

Q#	Quality Questions	Yes	Partly	No
Q1	Are the research aims clearly defined?			
Q2	Is the data collection procedure clearly described?			
Q3	Is the data pre-processing procedure clearly defined?			
Q4	Are the characteristics of the input data clearly described?			
Q5	Are the ML techniques well-defined?			
Q6	Is the training procedure clearly defined?			
Q7	Are the results and findings clearly stated?			
Q8	Is the proposed method compared to any statistical method?			
Q9	Is the proposed method compared to any other state-of-art ML method?			
Q10	Are the limitations of the study specified?			

**Table 4 bioengineering-10-00397-t004:** The ML methods used in the publications as well as the number of papers in each category and a brief description of each method.

Categories	Specific Methods	Description	Studies, *n*	References
Explicit approach	Random Forest Regression	A random forest is an ensemble learning approach built from many decision trees [23]. Ground truth is estimated through statistical harmonisation methods such as Combat [24].	1	[25]
	Convolutional neural networks (CNNs)	CNNs directly learn the mapping between intra-subject paired scans from two domains. Multiple layers of convolutional layers are used to gather and learn data features [26].	3	[27,28,29]
	Sparse dictionary learning	Models learn to create an implicit linear mapping between scanners based on dictionary representation [30]. Scanner-specific dictionary information can be learnt from a set of data and used for harmonisation.	1	[31]
	Transformer networks	Neural networks using transformer modules, which have spatial and/or image-level transformation capabilities, can learn shape and/or appearance differences across MRI domains [32].	1	[33]
	Generative adversarial networks (GANs)	GANs perform unsupervised learning to transform images [34]. Generators and discriminators work together and are against each other to convert MR images from source groups to a reference and vice versa.	7	[35,36,37,38,39,40,41]
	Autoencoders	It has an encoder-decoder architecture, where the MRI data is encoded into disentangled latent space structural and site-specific information and then decoded by harmonising the intensities with the embeddings without altering the structures [42].	7	[43,44,45,46,47,48,49]
Implicit approach	Adversarial Transfer Learning	This method indicates developing a learning system that composes of domain discriminators and feature extractors that focuses on the scanner invariant features while simultaneously maintaining performance on the main task of interest [50].	8	[51,52,53,54,55,56,57,58]
	Feature extraction-based transfer learning	The feature extraction modules network aims to extract a set of common features in each scanner/site and then map them to a gold-standard space to improve the performance of the final learning task [59].	3	[60,61,62]
	Fine-tuning	This approach utilises a well-trained model on a source dataset as the base and uses a small subset from the target dataset to re-train the model by updating the weight of layers in the re-trained model during the re-training process [63].	10	[64,65,66,67,68,69,70,71,72,73]
	Multi-task learning	Multi-task learning considers the site a task and learns the site-shared and site-specific features to generate more accurate models on multiple sites by assuming that the feature weights of different sites share similar sparse patterns [74].	1	[75]

**Table 5 bioengineering-10-00397-t005:** Datasets containing multiple scans of the same subject were used in the reviewed papers.

Data Repository	Related Study	Web Page if Applicable	Image Modality	Data Description
Zhejiang University Travelling Adults Dataset	[38]	https://brain.labsolver.org/test_retest.html, accessed on 15 February 2023	T1-weighted (sMRI), dMRI	3 travelling subjects were scanned in 10 scanners with different protocols within 13 months.
Multi-shell Diffusion MRI Harmonisation Challenge (MUSHAC) [76]	[31]	https://www.cardiff.ac.uk/cardiff-university-brain-research-imaging-centre/research/projects/cross-scanner-and-cross-protocol-diffusion-MRI-data-harmonisation, accessed on 15 February 2023	dMRI	14 healthy controls were scanned on three scanners with five acquisition protocols.
Human Connectome Project [77]	[39]	https://www.humanconnectome.org/study/hcp-young-adult/article/reprocessed-7t-fmri-data-released-other-updates, accessed on 15 February 2023	dMRI, Resting-state fMRI, Task-based fMRI	184 subjects which were scanned on a 3T and a 7T MRI scanner, separately.
SRPBS Travelling Subject MRI Dataset [78]	[46]	https://bicr-resource.atr.jp/srpbsts/, accessed on 15 February 2023	T1-weighted (sMRI), Resting-state fMRI	411 scans of the 3T MRI imaging data from 9 travelling subjects collected at 9 sites.
The Open Access Series of Imaging Studies (OASIS) 3 [79]	[44]	https://www.oasis-brains.org/, accessed on 15 February 2023	T1-weighted, T2-weighted (sMRI), resting-state fMRI, dMRI, etc.	A large longitudinal neuroimaging dataset that contains longitudinal scans with small intervals between different visits.
Private Dataset	[45]	-	T1-weighted (sMRI)	18 subjects were scanned on 4 different 3T scanners. The scans are at most four months apart.
Private dataset	[44]	-	T1-weighted FLAIR PD-/T2-weighted (sMRI)	12 subjects were scanned twice within 30 days on two scanners
Private dataset	[28]	-	dMRI	5 subjects were scanned using four scanners with different protocols.
Private dataset	[35]	-	T2, T2-FLAIR, T1-FLAIR	10 subjects were scanned using two scanners with 6 different protocols.

## Data Availability

Not applicable.

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
