# Peer review of "Machine Learning for Brain MRI Data Harmonisation: A Systematic Review"

_bioengineering, 2023, doi:10.3390/bioengineering10040397_

Round 1

Reviewer 1 Report

More motivation is required in the introduction section. 

A major contribution is not present in any of the section 

Table 3: Quality assessment questions and their answer is not available. How are these research questions helpful for brain MRI Data Harmonization?

Which SOTA approach the author has studied ? 

The author has suggested taking at least 5 projects else 5 MRI datasets and showcasing their facts and findings. as well as comparing their findings with the present and elaborating it .

Reviewer 2 Report

2.      The paper is long. You need to summarize some Sections and shorten the paper accordingly.

The paper presentation is not clear. What is the objective of the conducted research? What is the novelty related scientific literature ?

3.      The quality of the block diagrams and charts is poor. Please draw again

4.      Some paragraphs are too long. Please divide them into several short paragraphs to improve the readability.

5.      Enhance the readability of the paper, in particular, transitions from section to section should be smoother.

6.      The abstract is too general and not prepared objectively. It should briefly highlight the paper's novelty as what is the main problem, how has it been resolved and where the novelty lies?

7.      The related work section should be elaborated by including more papers and clear explanation of existing methodologies.

8.      More experiments, especially comparative experiments, should be involved to verify the performances of the proposed algorithm.

Reviewer 3 Report

This paper provides a review of ML algorithms for harmonizing MR brain images, including both explicit and implicit harmonization. The paper is generally easy to follow. Some comments are listed as follows:

1. The title is AI for brain MRI data harmonization. Probably the word AI is too big because only the ML methods are reviewed in the manuscript.

2. Some experimental analysis for ML performance would be helpful to understand state-of-the-art models.

3. Except for the quality assessment of existing research, the quality for MRIs is also important. Therefore, it is suggested to review some quality evaluation works in medical images, including RTN: Reinforced transformer network for coronary CT angiography vessel-level image quality assessment, A machine-learning framework for automatic reference-free quality assessment in MRI, etc.

4. The explanations of source and reference domains could be given more.

Round 2

Reviewer 1 Report

The author has suggested taking at least 5 projects else 5 MRI datasets and showcasing their facts and findings. as well as comparing their findings with the present and elaborating it . This question was raised but not addressed .

Author Response

Dear reviewer,

I would like to express my sincere gratitude for your valuable feedback on our manuscript. Your comments have been extremely helpful in improving the quality of our work, and we are grateful for your efforts in helping us to strengthen our paper.

Regarding your suggestion about showcasing at least five projects or five MRI datasets and comparing their findings, we agree that a summary of the current studies and a comprehensive comparative analysis would be highly beneficial. We acknowledge in the limitation section of our paper that we did not evaluate the performance of ML models or compare them with other models due to a lack of consistent evaluation metrics. In response to your suggestion, we have added a comparative analysis section to the discussion to summarize the findings in the comparative experiments conducted in all included studies. We hope this addition will address your concerns and provide more insight into the performance of different models.

We also want to highlight that we have analysed more than 40 papers in our review and summarized their findings in terms of methodology, datasets, models etc, in the supplementary materials. Additionally, we have provided a summary of all the datasets that include travelling subject data in terms of size, modality, and short summaries. Furthermore, we have showcased findings from studies that conducted comparative analysis in the discussion section. We believe that our efforts will address your concerns and provide a comprehensive review of the existing literature.

Besides that, we also improved our paper's referencing and general English writing. We hope that you can find the overall readability better!

Once again, we appreciate your feedback and hope that our revised manuscript meets your expectations and improves the overall quality of our work.

Sincerely,

Grace Wen

Reviewer 2 Report

The author adressed all the required modification

Author Response

Dear reviewer,

Thank you very much for reviewing our manuscript and providing us with such valuable feedback. We are pleased to hear that our revisions have successfully addressed your issues.

We appreciate the opportunity to improve our work based on your comments and suggestions, and we are grateful for the support you have given us throughout this process.

Best regards,
Grace Wen

Reviewer 3 Report

Thanks the authors for addressing my comments. One minor point is that except for MSE, SSIM etc, the specific metrics for medical images can be reviewed and better compared (see my previous comment).

Author Response

Dear reviewer,

 Thank you for your insightful feedback on our paper. We have carefully considered your suggestions and have made necessary revisions to improve the quality of our manuscript. We appreciate your efforts in helping us to strengthen our work.

Regarding your concern that we did not give sufficient attention to image quality assessment, we agree that this is an essential aspect of the research, and we appreciate the valuable insights. In our revised manuscript, we have expanded on this topic and provided additional insights on the current state of image quality assessment in MRI studies. In addition, referencing and general writing of the revised manuscript should be better.

In the discussion part, we emphasize that while harmonization should improve the quality of MRI images, evaluating the harmonized results is an essential component of image quality assessment. However, current studies do not have a consistent approach to assessing image quality and primarily use full-reference metrics such as signal-to-noise ratio and contrast-to-noise ratio. Moreover, reference-free IQA methods are lacking, and researchers often resort to group analysis (group-wise evaluation methods) when there are no references for comparison. We suggest that researchers consider using more advanced IQA or ML-based methods to address this limitation.

Furthermore, we recommend the development of a reliable and robust reference-free IQA framework for harmonization that can evaluate different MRI modalities and sequential stages. Such a framework should assess image quality at different levels, such as slice-wise, volume-wise, and subject-wise.  We argue that a thorough quality check is crucial before analysing the harmonised image.

Thank you again for your feedback, and we hope our revised manuscript addresses your concerns and improves the overall quality of our work.

Sincerely,

Grace Wen